# Secular-Trend Analysis of the Incidence Rate of Lung Squamous Cell Carcinoma in Taiwan

**DOI:** 10.3390/ijerph20021614

**Published:** 2023-01-16

**Authors:** Xiao-Han Shen, Yung-Yueh Chang, Rong-Qi Pham, Wei-An Chen, Fang-Yu Li, Wan-Chin Huang, Yu-Wen Lin

**Affiliations:** 1Master Program of Big Data in Biomedicine, College of Medicine, Fu Jen Catholic University, No. 510, Zhongzheng Rd., Xinzhuang Dist., New Taipei City 24205, Taiwan; 2Institute of Epidemiology and Preventive Medicine, College of Public Health, National Taiwan University, No. 17, Xu-Zhou Rd., Taipei City 10055, Taiwan; 3Department of Public Health, College of Medicine, Fu Jen Catholic University, No. 510, Zhongzheng Rd., Xinzhuang Dist., New Taipei City 24205, Taiwan; 4Data Science Center, College of Medicine, Fu Jen Catholic University, No. 510 Zhongzheng Rd., Xinzhuang Dist., New Taipei City 24205, Taiwan

**Keywords:** squamous cell lung carcinoma, incidence, age-period-cohort analysis

## Abstract

Lung cancer is the leading cause of cancer deaths worldwide, and squamous cell carcinoma (SQC) is Taiwan’s second most common lung carcinoma histotype. This study aimed to investigate changes in the long-term trend of the SQC incidence rate in Taiwan. SQC cases between 1985 and 2019 were adopted from Taiwan‘s Cancer Registry System; the age-adjusted incidence rate was calculated using the World Standard Population in 2000. The long-term trends of the age, period, and birth cohort effect of SQC incidence rates were estimated using the SEER Age-Period-Cohort Web Tool. The results revealed that the incidence of lung carcinoma in Taiwan increased, while the incidence of SQC exhibited a slight decrease during this study period. The age rate ratio (ARR) of the incidence rate in men declined gradually, and the period effect changed more slowly for women than men. The cohort effect formed a bimodal curve. The annual percentage change results for women indicated that the ARR decreased from 1.652 (95% confidence interval (CI): 1.422, 1.9192) at 30 to 34 years to 0.559 (95% CI: 0.4988, 0.6265) at 75 to 79 years; the period effect decreased from 1.2204 (95% CI: 1.1148, 1.336) in 1995 to 1999 to 0.608 (95% CI: 0.5515, 0.6704) in 2015 to 2019, with a greater decline in the later period. The cohort effect was unimodal, with the SQC risk value peaking in the 1915 birth cohort and exhibiting a steady decline thereafter. The results of this study suggest that a decrease in the smoking rate may be the reason for the decline in the incidence of SQC, and we observed a similar trend between SQC and the smoking rate in men.

## 1. Introduction

According to the GLOBOCAN 2020 data of cancer incidence and mortality estimates, lung carcinoma was the second most commonly diagnosed cancer and the leading cause of cancer death globally [1]. The global age-adjusted incidence rate of lung carcinoma is 31.5 per 100,000 for males and 14.6 per 100,000 for females [1]. According to the lung carcinoma study by Zhang et al. on the multi-country long-term trend between 1978 and 2012, the age-adjusted incidence rates for men in most countries decreased significantly, whereas that for women increased significantly [2]. Another study by Zhang et al. [3] reported that the age-adjusted incidence rates of lung carcinoma decreased significantly in Danish, Finnish, Swedish, English, German, Swiss, Italian, Polish, Canadian, African American, Caucasian American, Australian, and New Zealand males from 1973 to 2007. In contrast, the age-adjusted incidence of lung carcinoma increased significantly in females in these ethnic groups during the study period. From 1986 to 2017, the lung carcinoma age-adjusted incidence rate of both sexes in Taiwan showed an increasing trend, with that of males and females increasing from 22.5 to 43.5 and from 9.5 to 31.6, respectively, per 100,000 population [4]. According to the 2019 Taiwan Cancer Registry (TCR) annual report [5], lung squamous cell carcinoma (SQC) accounted for 18.89% and 3.55% of all lung carcinoma histotypes in males and females, respectively.

The results of age-period-cohort analyses of lung carcinoma incidence rates in Hong Kong [6], Shanghai [7], and India [8] revealed that the trend of the age-adjusted incidence rate was declining in Hong Kong but rising in Shanghai and India. Lortet-Tieulent et al. [9] analyzed the trend of squamous cell carcinoma (SQC) incidence rates in 11 countries from 1973 to 2002 in men and women aged 35 to 74 years, revealing that the incidence rate of SQC in men exhibited an overall decreasing trend for most countries except Spain and Iceland, whereas that in women exhibited an overall increasing or stabilizing trend. The annual percentage change (APC) of the age-adjusted incidence rates of SQC in American men and women from 2004 to 2009 were −0.5% and 1.7%, respectively [10]. The trends of the age-adjusted incidence rate of SQC for both sexes in Hong Kong from 1983 to 2000 decreased, with an APC of −3.8% for men and −7.6% for women [11]. The age-adjusted incidence rate of SQC in Beijing, China, from 2000 to 2016 peaked in 2007 with an APC of −2.6% for men. For women, the rate began to decline after peaking in 2004, and the APC was −5.4%, reflecting a greater decline for women than for men [12]. From 1995 to 2019, the incidence rates of SQC decreased from 9.29 to 8.46 per 100,000 population for men and decreased from 2.07 to 1.18 per 100,000 population for women in Taiwan [13]. The incidence rate peaked between 1999 and 2000 for both sexes; the overall changing incidence trend was characterized by an increase followed by a decrease. Compared with most countries, men exhibited a decreasing trend, but the decreasing trend was greater in Taiwan. Furthermore, the incidence trend was similar to that of Asian countries but different from that of Western countries.

The lung cancer incidence rate increase is attributed to adenocarcinoma in Taiwan. However, the trend of SQC, the second most common lung carcinoma histology, is opposite to adenocarcinoma. Therefore, we aimed to explore the factors that may influence the trend of SQC. Studies on the long-term trends of SQC incidence in Asia are limited. Thus, this study used the age-period-cohort analysis to analyze the characteristics of age, period, and birth cohort in the long-term trend of SQC incidence rates in both sexes and to explore the possible influencing factors.

## 2. Materials and Methods

### 2.1. Study Population and Data Sources

Data from 1985 to 2019 were obtained from the TCR. Lung carcinoma was indicated by the codes International Classification of Diseases for Oncology, Field Trial Edition (ICD-O-FT) T-162, and International Classification of Diseases for Oncology, Third Edition (ICD-O-3) C33-C34, and 279,166 cases were recorded during the study period. SQC was indicated by the morphology codes M8051/3, M8052/3, M8070/3, M8071/3, M8072/3, M8073/3, M8074/3, M8075/3, M8076/3, M8083/3, and M8084/3. The codes M8050/3 and M8078/3 were excluded. As a result, only 72 cases were registered under M8050/3, classified into other codes, and no cases in M8078/3. The number of SQC cases recorded was 52,921. Because the development of SQC before 30 years of age is rare, and because the classification of causes of death over 79 years of age is imprecise, we only included patients with new-onset SQC aged 30 to 79 years in the APC analysis. The total cases were 44,266. However, all ages were used to calculate the age-adjusted incidence rate. The period-specific, sex-specific, and age-adjusted population data were extracted from the Taiwan-Fujian Area Population Statistics of the Ministry of the Interior [14].

Since 1979, Taiwan has used the population-based TCR system, in which hospitals with more than 50 beds are used as the reporting unit. As a result, the overall percentage of morphologically verified cases (MV%) of lung carcinoma was 96.88%, and the percentage of death certificate only cases (DCO%) was 1.37% in 2019 [5].

### 2.2. Data Processing

The age-adjusted incidence was calculated using all ages on an annual basis. For the APC analysis, the patients aged 30 to 79 years old were divided into ten groups, namely those aged 30 to 34, 35 to 39, 40 to 44, 45 to 49, 50 to 54, 55 to 59, 60 to 64, 65 to 69, 70 to 74, and 75 to 79 years. The period from 1985 to 2019 was divided into groups of every five years, resulting in a total of seven groups, namely 1985 to 1989, 1990 to 1994, 1995 to 1999, 2000 to 2004, 2005 to 2009, 2010 to 2014, and 2015 to 2019. The birth cohort was calculated by subtracting the age group midpoint from the period midpoint of each group.

### 2.3. Statistical Analysis

This study used the World Standard Population Count of the year 2000 to calculate the age-adjusted incidence for both genders and the Age-Period-Cohort Web Tool developed by the Surveillance, Epidemiology, and End Results program (SEER) of the US National Cancer Institute for the analyses (https://analysistools.cancer.gov/apc/ accessed on 25 July 2022) [15]. The instrument is a log-linear Poisson regression tool, with the significance level set as 0.05; the Wald test was used for the two-tailed test. Long-term trend indicators included the average annual percentage change (AAPC), the annual percentage change (APC), the age rate ratio (ARR), the period rate ratio (PRR), and the cohort rate ratio (CRR).

The ARR was the indicator of the quantified influence of AAPC on age-associated natural history, accounting for the age effect; the PRR was the age-adjusted and cohort-adjusted period effect, with 2000 to 2004 adopted as the reference period; and the CRR was the age-adjusted and period-adjusted cohort effect, with the 1950 birth cohort used as the reference birth cohort. The chosen reference groups were based on the precision of the parameter estimates.

## 3. Results

Figure 1 illustrates the age-adjusted incidence rate of lung carcinoma and SQC in both sexes. The age-adjusted incidence rate of SQC in men per 100,000 population increased from 6 in 1985 to 12 in 2000 and then decreased to 8 in 2019; the fluctuation of the age-adjusted incidence rates of SQC in women remained at 1–2 per 100,000 population. The age-adjusted incidence rate of lung carcinoma exhibited an increasing trend; the AAPC of lung carcinoma in men was 2.2097 (95% confidence interval (CI): 2.0917, 2.3278), whereas the AAPC of lung carcinoma in women was 3.892 (95% CI: 3.6711, 4.1134). The SQC exhibited a slightly decreasing trend. The AAPC for SQC in men was −0.272 (95% CI: −0.5298, −0.0135), and the AAPC for SQC in women was −2.3791 (95% CI: −2.753, −2.0038). The incidence of SQC in men exhibited an upward trend before 2000 and began declining. For women, the incidence of SQC remained stable.

Figure 2 depicts the age-specific incidence of SQC from 1985 to 2019, which indicates that, in different periods, the incidence increased with age and decreased more rapidly in men than in women. Taking 2000 to 2004 as an example, the incidence in men per 100,000 population was 0.28 at the age of 30 to 34 years and reached 136 at the age of 75 to 79 years; the incidences for women in these age groups were 0.2 and 18, respectively, per 100,000 population. Table 1 presents the AAPC and APC for SQC in each age group for both sexes. The AAPC values revealed that the age-adjusted incidence declined more dramatically in women (−2.3791%) than in men (−0.272%). A comparison of the APCs of different age groups indicated that the incidence of men changed little for groups younger than the age of 75 to 79 years; the decrease for young women was more significant. The APC is 2.3699% (95% CI: 2.0465, 2.6944) for males and 1.2056% (95%CI: 0.3414, 2.0771) for females annually at the age group of 75 to 79 years.

Figure 3 illustrates the cohort-specific incidence rate of SQC from 1985 to 2019. The incidence rate for both sexes increased in the early cohorts and decreased in the late cohorts, with a slight difference noted between men and women. Compared with men, the decline of the cohort-specific SQC incidence in women was more pronounced. Taking the 75-to-79-year age group as an example, the incidence rate of SQC in men per 100,000 population rose from 52.7 in the 1910 birth cohort to 136.7 in the 1925 birth cohort and then declined to 109.8 in the 1940 birth cohort. For women per 100,000 population, SQC occurred at a rate of 6 in the 1910 birth cohort, rising to 18 in the 1925 birth cohort before decreasing to 11.5 in the 1940 birth cohort.

Figure 4 presents the analysis results of the APC of SQC in Taiwan from 1985 to 2019. As illustrated in Figure 4a, the ARR decreased with age in men, with a relative risk (RR) of 0.9317 (95% CI: 0.8499, 1.0215) for those aged 30 to 34 to 0.8243 for those aged 75 to 79 (95% CI: 0.7805, 0.8705). The estimation of the period effect revealed only slight changes for men, and the risk of SQC for men was 0.9361 (95% CI: 0.8814, 0.9943) in the period of 1985–1989. We observed a decreasing trend after the period of 1995–1999, and the risk value of SQC for men decreased to 0.8293 (95% CI: 0.7899, 0.8707) in the period of 2015–2019. We used the 1950 birth cohort as the reference group to estimate the cohort effect. The male group exhibited a double-peak curve, and the SQC risk for men showed an increasing trend from the 1910 to 1930 birth cohort, reaching its first peak of 1.33 in 1930 (95% CI: 1.2688, 1.3942), declining to a reversal point in 1960, increasing to reach a second peak in 1965, and then decreasing from that time onward.

Figure 4b presents the results of RRs for women. The ARR decreased slightly from 1.652 (95% CI: 1.422, 1.9192) in the age group of 30–34 years to 0.559 (95% CI: 0.4988, 0.6265) in the age group of 75–79 years. The period group 2000–2004 was the reference period group to estimate the period effect. The period effect for the female group exhibited an overall decreasing trend, decreasing from 1.2204 (95% CI: 1.1148, 1.336) in the period group 1995–1999 to 0.608 (95% CI: 0.5515, 0.6704) in the period group 2015–2019. To estimate the cohort effect, the birth cohort in 1950 was used as the reference birth cohort. The cohort effect for women was unimodal; the SQC cohort effect began to increase in 1910, reaching a peak of 1.7496 in 1925 (95% CI: 1.5471, 1.9787) and then exhibiting a steady downward trend after that. Except for the first cohort, which was nonsignificant, all the cohorts showed significant differences.

## 4. Discussion

The results of this study indicated that the incidence rates for both men and women in different periods increased with age and only decreased for the age group of 70–74 years in the period group of 1985–1989. We observed an upward trend for the cohort-specific incidence rates, regardless of age or sex, before the 1930 birth cohort, followed by a decreasing trend. Following adjustment for the period and cohort, the age risk (ARR) decreased for both men and women, with the effect more pronounced for women; the RR decreased from 0.93 to 0.82 and from 1.65 to 0.56 for men and women, respectively. The period effect began to decline after the period group of 1990–1994 for both men and women, and the decline was more obvious for women. The RR decreased from 0.91 to 0.79 and from 1.12 to 0.67 for men and women, respectively. The cohort effect peaked in the 1930 and 1965 birth cohorts for men, with an RR of 1.33 and 1.18, respectively. For women, the cohort effect peaked in the 1925 birth cohort, with an RR of 1.75.

In the studies of SQC incidence rates in Hong Kong from 1983–2000 [11], Beijing from 2000–2016 [12], and Japan from 1975–2003 [16], an overall decreasing trend was observed; however, the periods in which the decreases happened were different in each region. The incidence rates of SQC in Beijing peaked in 2007 for men and in 2003 for women, whereas in Osaka, Japan, these rates peaked between 1989 and 1993 for men and between 1984 and 1988 for women. The period of this study was between 1985 and 2019, and the peak periods for men and women were the period groups of 2000–2004 and 1995–1999, respectively. Different studies have reported that women reach the peak period earlier than men. However, because the study periods and comparison benchmarks were different, the periods in which the peaks occur also differ.

Lortet-Tieulent et al. [9] divided seven countries’ lung carcinoma incidence rates based on histology and conducted an age-period-cohort analysis of the incidence trends accordingly. The analysis revealed that from 1910 to 1960, the overall trend of the SQC cohort effect risk among men decreased. The only exception was the SQC cohort effect risk for men in Spain, which increased with the birth cohort. On the other hand, the overall trend for women mainly was an increasing trend followed by a decreasing trend, except in France, where the trend continues to rise, and in Spain, where the trend first decreased and then increased. Compared with the cohort effect results of Lortet-Tieulent et al., the trend for men in Taiwan differed from that in most other countries, presenting a double-peak curve. By contrast, the trend for women was similar to that of most countries in Lortet-Tieulent et al., beginning with an increasing trend followed by a decreasing trend.

Smoking is the leading risk factor for lung carcinoma [17,18,19,20]. Approximately 85% of lung carcinoma can be attributed to smoking [21,22], and smoking is also related to SQC and small cell lung carcinoma (SCLC) [22]. Therefore, we plotted the age-adjusted incidence rates of SQC from 1979 to 2019 and the smoking rates [23], as obtained from the Health Promotion Administration, Ministry of Health and Welfare survey of Taiwanese people aged over 18 years from 1990 to 2014 (Figure 5), to observe the changing trends of these two variables. As presented in Figure 5, the age-adjusted incidence rates of SQC in Taiwan peaked in 2000 and 1999 for men and women, respectively. Then, we observed a decline for both sexes, with a greater decline for men. The overall smoking rate for men decreased, from 59.4% in 1990 to 29.2% in 2014, whereas that for women fluctuated slightly. In addition, a study revealed that cigarette sales declined sharply when tobacco taxes were raised [24]. Therefore, the change in the smoking rate may be related to the Tobacco Hazards Prevention Act implemented by the Taiwanese government in 1997 and the introduction of the Tobacco Health Welfare Tax in 2002, both of which were explicitly introduced to reduce the smoking rate. In 2006 and 2009, the Tobacco Health Welfare Tax on tobacco products was increased.

The smoking rate in Taiwan has continued to decline since 1990, and the SQC incidence rate in men began to decline in 2000. According to Weiss’ estimation, the latency period from smoking to lung carcinoma mortality in the population is approximately 30 years [25], which is different from the trend we observed. The smoking rate may have begun to decrease before the implementation of the Tobacco Hazards Prevention Act. However, we estimated that the incidence rate of SQC in men will continue to decline. The incidence of SQC in women is less likely to change relative to the smoking rate. Implementing a tobacco control program in California was associated with a reduced incidence of lung carcinoma. The study results of Barnoya and Glantz indicated that the tobacco control program was estimated to lead to a decrease of approximately 6% in lung carcinoma cases [26]. The increasing incidence of lung carcinoma in different countries around the world also parallels the trend of cigarette consumption [21], and the change in the mortality of lung carcinoma in the United States also parallels the trend of cigarette smoking [17]. A study also revealed that in 1980, the incidence rate of SQC in the United States decreased in parallel with the smoking rate [27]. Another survey of cigarette consumption and lung carcinoma mortality in men in the United States indicated that the change in mortality was associated with decreased cigarette consumption. Whittemore used simple linear regression to estimate the effect of cigarette smoking on mortality and determined that cigarette consumption explained 93% of the variation in mortality [28].

Smoking has a dose-dependent relationship with the risk of lung carcinoma. The risk is reduced following smoking cessation, especially for those who quit early [21], indicating that smoking is an avoidable and preventable risk factor. Therefore, to date, the implementation of the Tobacco Hazards Prevention Act has been effective. Still, the impact of smoking on lung carcinoma incidence in Taiwan and other Asian countries is not as significant as that in Western countries. Jung et al. proposed two possible reasons for this phenomenon. First, the per capita consumption of cigarettes in Western countries reached a certain amount in an earlier period than in Asian countries. The effect of smoking on lung carcinoma may thus not have been revealed in Asian countries. Second, people in Asian countries start smoking later than those in Western countries. However, for the later birth cohort, the age at which they start smoking declined gradually [29]. Therefore, the double peak we observed in the male cohort (Figure 4) may be related to the smoking population comprising increasingly younger people.

In contrast to smokers, a proportion of lung carcinoma in nonsmokers is caused by secondhand smoke [21], and exposure to secondhand smoke at home during adulthood leads to a significantly increased risk of developing lung carcinoma [30]. Among several studies of lung carcinoma risk factors among women [20,31,32,33], two studies published by Ko et al. [31,32] suggested that cooking fumes may be a risk factor for lung carcinoma in Taiwanese nonsmoking women. Cooking fumes significantly increased the risk of lung carcinoma (odds ratio (OR) = 2.5 (1.4, 4.3)), and the study results indicated a dose–response trend between the number of meals cooked per day and the risk of lung carcinoma. In addition to cooking fumes, a study by Chen et al. [34] revealed that the APOBEC mutation signature and carcinogen exposure may be early drivers of lung carcinoma in female nonsmoking patients with lung carcinoma; additionally, relevant studies have revealed that the TP53 gene mutation can be observed in almost all patients with SQC [35,36], and this gene mutation may affect the incidence of SQC in Taiwan.

In relevant studies of the association between diet and lung carcinoma risk, cruciferous vegetables [37,38] and soybeans [39] have been revealed to be potentially protective factors against lung carcinoma. Soybean intake was significantly and negatively associated with lung carcinoma incidence in women (RR = 0.79 (0.67, 0.93)), nonsmokers (RR = 0.62 (0.51, 0.76)), and Asian populations (RR = 0.86 (0.74, 0.98)) [39]; the dietary intake of micronutrients may also be associated with lung carcinoma risk [40,41]. Magnesium intake was positively associated with lung carcinoma risk in men and current smokers, and increased dietary calcium intake had a protective effect only in women, reducing the risk of lung carcinoma by 11% to 18% [40]. In nonsmoking women with a low calcium intake, calcium and phosphorus were negatively associated with lung carcinoma risk (calcium: hazard ratio (HR) = 0.66 (0.48, 0.91); phosphorus: HR = 0.55 (0.36, 0.85)) [41]. In addition, the vitamin B6 level in serum is negatively associated with the risk of lung carcinoma (OR = 0.44 (0.33, 0.60)) [42]. According to the study of Büchner et al. [43], the diversified consumption of fruits and vegetables alone by current smokers reduces the risk of SQC (vegetables: HR = 0.88 (0.82,0.95); fruits: HR = 0.84 (0.72, 0.97)). Pan et al. [44] compared the 1993–1996 and 2005–2008 Nutrition and Health Surveys in Taiwan, revealing that the intake of fruits, vegetables, and legumes have increased for both sexes. Furthermore, the downward trend of SQC indicated that the intake of fruits, vegetables, and legumes might be one of the factors driving the decline in SQC incidence [41]. A comparison of the 2013–2016 and 2017–2020 Nutrition and Health Surveys in Taiwan [45,46] revealed that vegetable and fruit intake for both sexes increased with age, which may have an impact on the age effect of the incidence of SQC in the future. Regarding the intake of trace elements, the average intake of calcium, phosphorus, and vitamin B6 increased among both men and women, and the average intake of calcium increased with age for both sexes. The average iron intake in women has increased slightly in recent years and may affect the incidence of lung carcinoma in Taiwan.

The strength of this study is that it complements the analysis of long-term trends of the incidence rates of SQC in Asia by using population-wide incidence data, which are highly representative. Furthermore, the study period was almost 35 years, and the smoking rate data are available for basic validation and the inference of possible exogenous factors. However, this study has some limitations. This study employed population data but needed more data on individuals and risk factors, thus potentially generating ecological fallacies. Furthermore, the incidence of SQC may be underestimated because of possible gaps in cancer registry data from 1985 to 1994 and for the age of 30–70. However, this would not affect the results of this study. Most studies have used the APC model to perform the time-trend analysis of lung carcinoma incidence; trend studies based on histological types are lacking. In the present study, we analyzed the long-term trend of SQC incidence according to sex, age, period, and birth cohort. We used population data to explore the possible correlations between risk factors and SQC. The results provide a basis for examining the underlying causal mechanism and can be used as a reference for future research.

## 5. Conclusions

The study revealed that the age, period, and cohort effects were associated with the incidence rates of SQC in both men and women. The ARR and the risk of period effects decreased with age for both sexes. This decreasing trend was attributable to the implementation of policies aimed at reducing smoking, and the increased Tobacco Health Welfare Tax on tobacco products affected the age effect. The effect of the birth cohort on the incidence rate was the largest among all the effects, and the RR exhibited an overall decreasing trend. The peak for men was in the 1930 and 1965 birth cohorts, and that for women was in the 1925 birth cohort. This result is presumably related to the age at which people start smoking and policy intervention. In this study, we noted a similar trend between SQC and the smoking rate in men. The study results provide evidence for the reinforcement of antismoking measures to reduce the incidence of SQC in Taiwan. Therefore, long-term trend monitoring of the relationship between smoking and lung carcinoma must be continued in the future.

## Figures and Tables

**Figure 1 ijerph-20-01614-f001:**
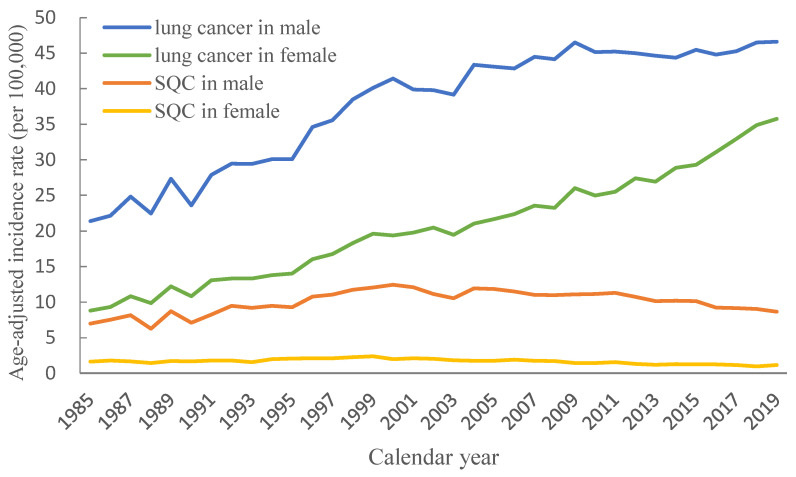
Age-adjusted incidence rates of lung cancer and squamous cell lung carcinoma stratified by sex in Taiwan between 1985 and 2019.

**Figure 2 ijerph-20-01614-f002:**
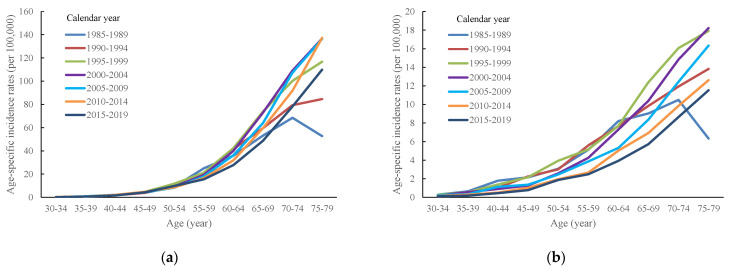
Age-specific incidence rates of squamous cell lung carcinoma in Taiwan between 1985 and 2019 by calendar periods and age groups: (**a**) men and (**b**) women.

**Figure 3 ijerph-20-01614-f003:**
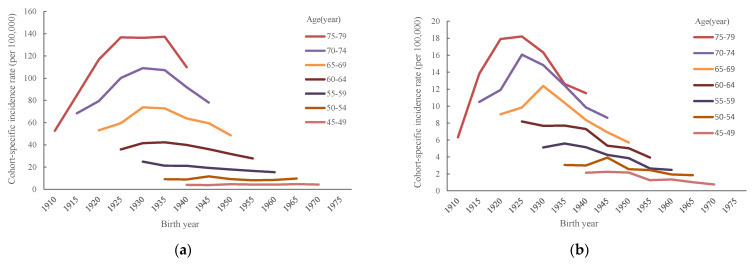
Cohort-specific incidence rate of squamous cell lung carcinoma in Taiwan between 1985 and 2019 by age groups: (**a**) men and (**b**) women.

**Figure 4 ijerph-20-01614-f004:**
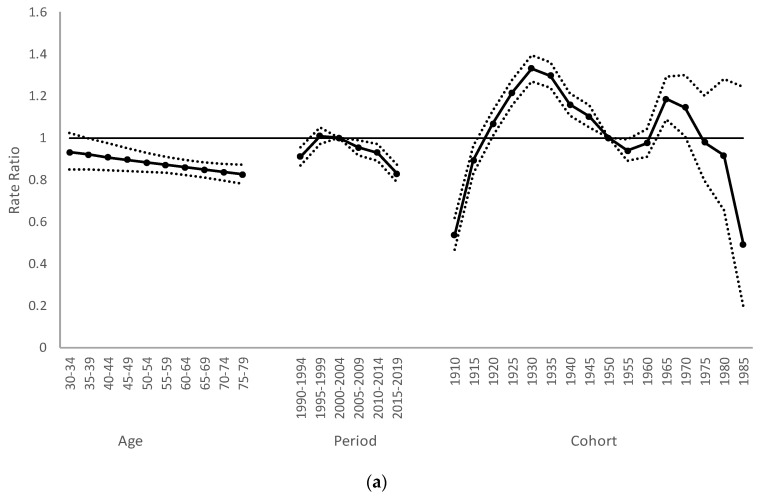
Age-period-cohort effect of squamous cell lung carcinoma: (**a**) men and (**b**) women.

**Figure 5 ijerph-20-01614-f005:**
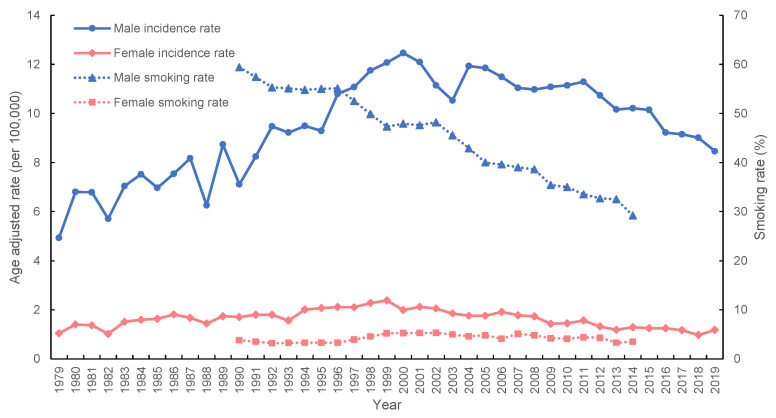
The squamous cell lung carcinoma (SQC) incidence rates between 1979 and 2019, and the smoking rate between 1990 and 2014 in Taiwan.

**Table 1 ijerph-20-01614-t001:** Net drift and local drift by age groups of squamous cell lung carcinoma in Taiwan between 1985 and 2019.

		Male			Female		
		%	95% CI		%	95% CI	
AAPC (net drift)		−0.272	−0.5298	−0.0135	−2.3791	−2.753	−2.0038
APC (local drift)	30–34 year	−1.5973	−3.6327	0.481	−3.5023	−5.5998	−1.3581
	35–39 year	−0.0177	−0.8377	0.8092	−3.651	−4.8823	−2.4038
	40–44 year	0.1028	−0.4145	0.6226	−3.3519	−4.1581	−2.5389
	45–49 year	0.0606	−0.2781	0.4005	−3.1537	−3.7618	−2.5418
	50–54 year	−0.5529	−0.7976	−0.3077	−2.5587	−3.0139	−2.1014
	55–59 year	−1.226	−1.4168	−1.0348	−2.76	−3.1566	−2.3617
	60–64 year	−1.0689	−1.2263	−0.9113	−2.4751	−2.8377	−2.1111
	65–69 year	−0.3756	−0.524	−0.2271	−1.7463	−2.1208	−1.3704
	70–74 year	0.6146	0.4397	0.7899	−0.9537	−1.4055	−0.4999
	75–79 year	2.3699	2.0465	2.6944	1.2056	0.3414	2.0771

AAPC: average annual percentage change; APC: annual percentage change; CI: confidence interval.

## Data Availability

Publicly available datasets were analyzed in this study. This data can be found here: https://cris.hpa.gov.tw/ accessed on 25 July 2022 (in Chinese).

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
