# Peer review of "Secular-Trend Analysis of the Incidence Rate of Lung Squamous Cell Carcinoma in Taiwan"

_ijerph, 2023, doi:10.3390/ijerph20021614_

Round 1

Reviewer 1 Report

Dear authors,

Congratulations for your present work. It is an interesting analysis of the trends of squamous lung cancer incidence in Taiwan. Introduction is comprehensive enough and Material and Methods are properly described and are appropriate for the objectives of the study. Results are clearly presented and Discussion/Conclusions are well substantiated with the results.

However, I have minor comments for the acceptance of your manuscript:

Comment 1:  Introduction, lines 34-36. According to Globocan 2020 Data, lung cancer was not the second most prevalent worldwide. Globocan 2020 did not show fatality rates but mortality rates. Referring to mortality, lung cancer presented the highest mortality rates worldwide. Please, revise Globocan data and consider modifying wording.

Comment 2: Introduction, line 63. Data from 1985-2019 is from Taiwan Cancer Registry?. Please, consider modifying wording to clarify this issue.

Comment 3: Materials and Methods, line 78. ICD-O-FT and ICD-O-3 classifications are mentioned for first time, they probably need to be cited entirety. I.e.  International Classification of Diseases…..

Comment 4: Materials and Methods, lines 79-81?. Why did you not include 8050/3 and 8078/3 codes?.  According to ICD-O-3 and IARC’s Cancer Incidence in Five Continents, they are squamous cell histologies. This issue need to be clarified at this point of the section.

Comment 5:  Materials and Methods, lines 81-83. Regarding to the exclusion of cases aged 80+ due to imprecision on cancer death certificates, maybe it would be more appropriate to exclude DCO cases instead of 80+ age group. I suggest that you should comment this issue in Discussion, as a limitation of your study.

Comment 6: Materials and Methods, line 112. Why did you choose 2000-2004 as reference period?. A brief explanation is needed at this point of Materials and Methods.

Comment 7: Materials and Methods, line 113. Why did you choose 1950 as the reference birth cohort?. A brief explanation is needed at this point of Materials and Methods.

Comment 8: Results, line 141. Figure 2 legend can be improved: Age-adjusted incidence rates of squamous cell lung carcinoma in Taiwan between 1985 1 and 2019 by calendar periods and age groups….

Comment 9: Results, line 143. Table 1 legend can be improved: Net drift and local drift by age groups of squamous cell lung carcinoma in Taiwan between 1985 and 2019..

Comment 10: Results, line 155. Figure 3 legend can be improved: Cohort-specific incidence rate of squamous cell lung carcinoma in Taiwan between 1985 and 2019 by age groups.

Comment 11: Conclusions, line 329. You used the terms ‘positive correlation’ in reference to your results in SQC incidence rate and the observed smoking rate in men in Taiwan. These terms can lead the reader to understand that you did performed a statistical test of correlation, but it was not in this way. I suggest you to modify the wording of this sentence.

Comment 12: References, line 363. Reference number 5 need to be improved in their wording: ie. Health Promotion Administration; Ministry of Health and Welfare. Taiwan. 2019 Cancer Registry Annual Report. Available online…….

Reviewer 2 Report

The analysis of lung cancer incidence is very relevant to understand the trends and future impact of the lung cancer in a particular country. The authors have done a good job describing the APC trends for squamous lung cancer using population based cancer registry data. The analysis is well done and interesting, and the discussion is systematic and covering most relevant issues. 

There is a major point I would like to suggest to the authors, namely, the need to complement the squamous trend with the adenocarcinoma trend. I found this point particularly relevant to understand the global trend of lung cancer. If I interpreted correctly the figure one, the trends for adeno and squamous are not similar.  This increases the usefulness of the comparative analysis of trends. Without this, I find the paper a partial report.

a couple of minor points: 

- title of section 2.3. Could it be 'statistical analysis' instead of statistic?

- Figure 2: could it be 'age-groups incidence rates' instead of age-adjusted'

Reviewer 3 Report

The manuscript entitled “Secular-trend analysis of the incidence rate of lung squamous cell carcinoma in Taiwan” by Shen et al. provides the long-term trends of squamous cell carcinoma (SQC) in Taiwan. Although the trends of SQC incidence rates in men and women are interesting for the country, a positive association between smoking and lung cancer has already been established. In this regard, the finding of this paper has a limited impact.  However, the authors claim that the study results provide evidence of the state’s antismoking reinforcement measures that reduced the incidence of SQC. Reporting such causal mechanisms of SQC is important as it may encourage policymakers to apply intervention measures to prevent disease at an early stage.

Below are some of my concerns and suggestions:

Female breast cancer is the most commonly diagnosed cancer followed by lung, however, lung cancer is still the leading cause of cancer death. Authors need to justify “lung cancer was the leading cause of cancer death worldwide” or correct the statement in the abstract.

Among the lung cancer subtypes, adenocarcinoma (ADC) is the most common than SQC. It would have been better to address why authors only preferred the specific subtype. Would there be a different trend if both ADC and SQC were combined? Does the Lung cancer cohort include all the subtypes?

There are some typos and/or grammatical errors.

Round 2

Reviewer 2 Report

Although I may accept that the analysis of the squamous LC in isolation is a reasonable option, I sill find this option reductive. Then, my suggestion was intended to offer a more complete view of the lung cancer trends in the country.